# Spatiotemporal lagging of predictors improves machine learning estimates of atmosphere–forest $CO_2$ exchange

Matti Kämäräinen[1], Juha-Pekka Tuovinen[2], Markku Kulmala[3], Ivan Mammarella[3], Juha Aalto[1,4], Henriikka Vekuri[2], Annalea Lohila[2,3], Anna Lintunen[3,5]

[1]Weather and Climate Change Impact Research, Finnish Meteorological Institute, Helsinki, Finland

[2]Climate System Research, Finnish Meteorological Institute, Helsinki, Finland

[3]Institute for Atmospheric and Earth System Research / Physics, Faculty of Science, University of Helsinki, Helsinki, Finland

[4]Department of Geosciences and Geography, University of Helsinki, Helsinki, Finland

[5]Institute for Atmospheric and Earth System Research / Forest Sciences, Faculty of Agriculture and Forestry, University of Helsinki, Helsinki, Finland

*Correspondence to*: Matti Kämäräinen (matti.kamarainen@fmi.fi)

**Abstract.** Accurate estimates of net ecosystem $CO_2$ exchange (NEE) would improve understanding of natural carbon sources
and sinks and their role in the regulation of global atmospheric carbon. In this work, we use and compare the random forest (RF) and the gradient boosting (GB) machine learning (ML) methods for predicting year-round 6 hourly NEE over 1996–2018 in a pine-dominated boreal forest in southern Finland and analyze  predictability of NEE. Additionally, aggregation to weekly NEE values was applied to get information about longer term behavior of the method. The meteorological ERA5 reanalysis variables were used as predictors. Spatial and temporal neighborhood (predictor lagging) was used to provide the
models more data to learn from, which was found to improve considerably the accuracy of both ML approaches compared to using only the nearest grid cell and time step. Both ML methods can explain temporal variability of NEE in the observational site of this study with meteorological predictors, but the GB method was more accurate. Only minor signs of overfitting could be detected for the GB algorithm when redundant variables were included. Accuracy of the approaches, measured mainly using cross-validated $R^2$-score between the model result and the observed NEE, was high, reaching a best estimate
value of 0.92 for GB and 0.88 for RF. In addition to the standard RF approach, we recommend using GB for modeling the $CO_2$ fluxes of the ecosystems due to its potential for better performance.

## 1 Introduction

Forests and other terrestrial carbon sinks remove about one third of the anthropogenic carbon dioxide ($CO_2$) annually emitted to atmosphere, and thus they constitute an important component of the global carbon balance (Friedlingstein et al., 2020). However, the existing observation network for estimating the total atmosphere–ecosystem $CO_2$ exchange is sparse (Alton, 2020), and especially historical coverage of observations over the past decades is poor. Among other biotypes and ecosystems, boreal forests contribute significantly to the global atmospheric carbon stock, but how they do it in detail is still largely unknown, reflected in the wide range of estimates of the carbon storage of these forests (Bradshaw and Warkentin, 2015). Therefore, there is a need for accurate spatio-temporal modeling of carbon fluxes for improved monitoring and understanding the boreal, and ultimately, the global carbon cycles (Jung et al., 2020).

In boreal forests, the atmosphere–ecosystem $CO_2$ flux shows strong seasonal and diurnal cycles, dominated by 1) the photosynthesis by plants (acting as a $CO_2$ sink from the atmosphere), and 2) by the total ecosystem respiration, including plant respiration and organic decomposition processes by microorganisms (acting as a $CO_2$ source into the atmosphere). In a homogeneous forest environment, the net flux generated by these processes can be accurately measured with the micrometeorological eddy covariance method, which has emerged as common standard for long-term ecosystem-scale flux measurements (Aubinet et al., 2012; Hicks and Baldocchi, 2020).

Both total respiration and photosynthesis are typically at their largest in the warm season in boreal forests (Ueyama et al., 2013; Wu et al., 2012; Kolari et al., 2007). On average, their net effect, i.e. net ecosystem exchange of $CO_2$ (NEE), is dominated by photosynthesis on the weekly scale in summer, but on sub-daily scale, the total respiration turns NEE positive during nights when photosynthesis of plants is switched off. In the cold season, diurnal variability is mostly absent, and then NEE is again slightly positive as respiration still dominates.

Various meteorological and local abiotic and biotic factors and processes affect NEE, and their importance is different in different seasons. Local conditions include soil type and properties, and plant species and their density distributions. Key meteorological variables, such as air temperature and short-wave radiation, typically have large seasonal and diurnal variations. These variables are observed globally using in-situ and remote sensing techniques, and the resulting large-scale data sets can be further post-processed and homogenized via data assimilation, employing numerical weather prediction (NWP) models, and presented in a spatiotemporal grid format. This product is called reanalysis, which can be considered a by-product of the NWP process (Parker, 2016).

In recent years, various machine learning (ML) approaches have been proposed and used to model NEE or related quantities

over various locations and globally (Jung et al., 2020; Besnard et al., 2019). Even though NEE appears to be a difficult quantity to model accurately (Tramontana et al., 2016), the random forest method (RF) has been shown to be suitable for this task (Shi et al., 2022; Nadal-Sala et al., 2021; Reitz et al., 2021; Tramontana et al., 2015; Zhou et al., 2019). Typically, the previous work has concentrated on modeling the rather coarse weekly, monthly, or annual temporal resolution: however, some exceptions with subdaily resolution exist (Bodesheim et al., 2018).


Here we employ the RF algorithm to model the 6 hourly NEE between the atmosphere and a boreal forest in Finland. In addition to the RF regression method, we use the gradient boosting (GB) regression (Friedman, 2001; Chapter 10 in Hastie et al., 2009), which has not been as common as the RF in this context. Both methods of this study fit an ensemble of regression trees to predict NEE. The potential of the GB resides in the fitting process: while the trees of RF are fit independent of each

other, GB trees become aware of the prediction error of the previous trees as the fitting process continues sequentially, allowing them to concentrate on the most difficult samples (Chen and Guestrin, 2016).

Several meteorological predictors from the global ERA5 reanalysis (Hersbach et al., 2020) were used as input for the RF and GB regression models, including but not limited to soil and air temperatures, precipitation amounts, radiation quantities, and

heat fluxes. In contrast to previous studies we use only the raw reanalysis quantities as predictor input: specifically, we do not use meteorological in-situ data directly (e.g. Mahabbati et al., 2021; Tramontana et al., 2015) nor satellite data directly (e.g. Zhou et al., 2019). By excluding many input data sources we can simplify considerably the modeling process. However, in-situ and satellite data have been used in the assimilation of the reanalysis itself (Hersbach et al., 2020).

We propose, test, and show the value of using spatiotemporal neighboring information from the ERA5 reanalysis for improving modeling results. Previously, temporal neighborhood has been used to improve the modeling; see, for instance, Besnard et al., 2019. Benefits of using the temporal neighborhood together with the spatial neighborhood has not been studied earlier.

Additionally, we investigate in detail whether the skill of the GB method could overcome the skill of the popular RF method in explaining the variability of NEE when using the meteorological predictors. For that, we first tune the hyperparameters of both ML methods carefully using Bayesian optimization (Snoek et al., 2012), and compare their results. Then, we rank the importance of the individual predictors in the study site using the SHAP analysis (Lundberg et al., 2020) and explore the effect of reducing both the number of samples and the number of predictors on the accuracy of the GB model. Finally, we

discuss the significance of our results in a broader context.

## 2 Materials and methods

### 2.1 CO2 flux measurements as the target variable

Eddy covariance $CO_2$ flux data, measured above a 60 year old Scots pine forest in Hyytiälä, Finland (61°51' N, 24°17' E) in 1996–2018, corresponding roughly a footprint area of 125 000 m² (Launiainen et al., 2022) and processed to represent NEE,
were acquired from https://smear.avaa.csc.fi/download (accessed 25 February 2021). Flux processing for NEE was done using the EddyUH software (Mammarella et al., 2016; a summary of the data is shown in Fig. 1, presented as multi-year mean values). NEE is a sum of ecosystem carbon uptake in photosynthesis and carbon loss in respiration, and a negative NEE means that the forest takes up carbon, i.e., is a carbon sink. These data consist of 30 min averages which were aggregated for modeling to 6 h resolution using averaging with moving, non-overlapping windows. For this, the 00, 06, 12,
and 18 hours were used: the hour values indicate the beginning of the averaging period. Only complete 6 hourly aggregates, i.e. those with no missing values arising from flux processing and instrument faults, were accepted for the averaging process. The resulting data set contained 10500 non-missing data points and 22800 missing values. In addition to the preprocessed NEE data, the modeling was separately tested using the raw $CO_2$ flux (i.e., measured by the eddy covariance system and without storage change flux correction and friction velocity filtering) as the target variable.


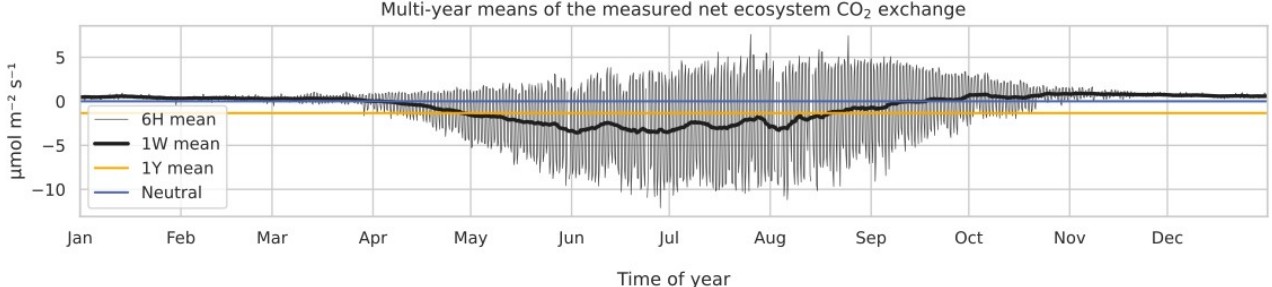

**Figure 1: The 6 hourly (thin black), weekly (thick black), and annual (orange) multi-year means of observed net ecosystem CO₂ exchange (NEE) at the Hyytiälä SMEARII site. Eddy covariance method with a 24-m tall tower was used for measurements. Years 1996–2018 were used in calculation of the mean values.**


Additionally, weekly means were calculated from the 6 h data for validation purposes. For this, a moving, overlapping and centered windowing was used to preserve the same number of samples as in the 6 h data. Missing data inside the window were accepted not to discard almost all of the samples. When validating the model, the missing 22800 time steps were also rejected from the model results for consistency. The diurnal distribution of the missing data was the following: 00 UTC –
75%; 06 UTC – 66%; 12 UTC – 59%; and 18 UTC – 74%.

## 2.2 Variables from the ERA5 reanalysis as predictors

Typically, air and soil temperatures, short-wave (photosynthetically active) radiation, and relative humidity are the key meteorological variables used in modeling the $CO_2$ flux (eg., Nadal-Sala et al., 2021). In addition to these, we included a large set of other variables 1) to search for new, unexpected relationships between the flux and these less common variables, and 2) to study how much these variables can either improve or deteriorate the accuracy of the model. Altogether 19 meteorological variables from the global ERA5 reanalysis product (Hersbach et al., 2020) were selected (Table 1).

The ERA5 reanalysis data for 1996–2018 were downloaded from https://cds.climate.copernicus.eu/ (accessed 15 March 2021) in the 1°×1° spatial and 1 h temporal resolution. The data were downsampled to 6 hourly using moving averaging with non-overlapping windows, following the same procedure as with the $CO_2$ flux data.

**Table 1. Gridded parameters from the ERA5 reanalysis product. Asterisks (*) indicate parameters which were found to be redundant – containing irrelevant or superfluous information compared to other parameters – and for this reason they were excluded from the final fitting of the model.**

| Variable | Abbreviation |
| --- | --- |
| Evaporation | e |
| Mean surface direct short-wave radiation flux | msdrswrf |
| Mean sea level pressure | msl* |
| Mean surface latent heat flux | mslhf |
| Mean surface sensible heat flux | msshf |
| Relative humidity at 1000 hPa | r |
| Snow depth | sd* |
| Soil temperature, level 1 (7 cm) | stl1 |
| Soil temperature, level 2 (28 cm) | stl2* |
| Soil temperature, level 3 (100 cm) | stl3 |
| Volumetric soil water, layer 1 (0–7 cm) | swvl1* |
| Volumetric soil water, layer 2 (7–28 cm) | swvl2* |
| Volumetric soil water, layer 3 (28–100 cm) | swvl3* |
| 2-meter temperature | t2m |
| Total cloud cover | tcc* |
| Total precipitation | tp* |
| 10-meter u-component of the wind | u10 |
| 10-meter v-component of the wind | v10 |
| Geopotential at 150 hPa | z* |

## 2.3 Temporal lagging and spatial neighbourhoods of the predictor data

As the first approximation, the modeling could be carried out by using the grid point closest to the Hyytiälä site. Similarly, temporal synchronization of the predictor data and the target variable could be used. On the other hand, many processes

happen sequentially in time and their effect on the target variable could be seen as delayed. For example, meteorological conditions at night-time can affect plant photosynthesis the following day (Kolari et al., 2007). On the other hand, because of biases and other uncertainties of the ERA5 reanalysis, the nearest grid cell might not represent the best estimate of the variability of different quantities – instead, the nearby cells could do that. We wanted to give the ML models the opportunity

to take these effects into account, and selected 25 closest grid cells around the site and five closest time steps around each of the time steps (t=0) of the target variable. Note that lagging was applied both to forward and delay the predictors in time (Fig. 2a). The reason to use also the negative lags, even though they seem to violate causality, are the potential temporal uncertainties of the reanalysis data.


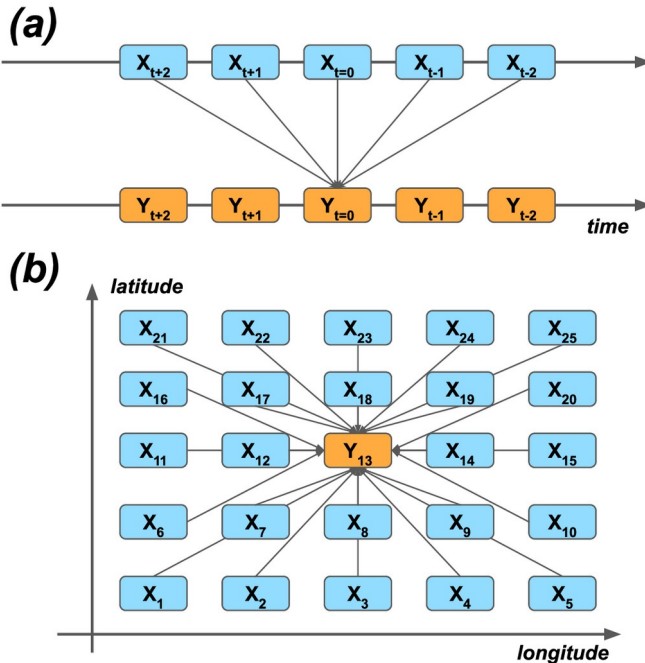

**Figure 2: The principle of using (a) temporal lagging and (b) spatial neighborhoods of predictor variables X to model the target variable Y. The numbering of the (a) lags and (b) grid cells corresponds the lags and neighbors of the ERA5 data used in the study.**

In total, we had 19 variables × 25 grid cells × 5 temporal lags = 2375 individual predictors for modeling. Technically, calculation of the correlation matrix was too laborious a task with $2375^2 \approx 5.6 \times 10^6$ operations. However, the predictor set necessarily contains highly correlated variables: for example, the temperature time series of neighbouring grid cells are correlated. Such collinearity can hamper the robustness and reliability of statistical models (Lavery et al., 2019). To deal

with the collinearity, the principal component analysis method (Jolliffe and Cadima, 2016) using 1) all components and 2)
reduced number of components was tested as a preprocessing step to make the predictors orthogonal, i.e., non-correlated, but
it was found that this dimension reduction method was unnecessary, as the results were slightly better without it (not shown),
and thus it was not used here.

### 2.4 Gradient boosting and random forest regressions
For the machine learning of this study, the xgboost package (version 1.4.2: https://xgboost.readthedocs.io/; Chen and
Guestrin, 2016) of the Python language (v. 3.7.6: https://www.python.org/) was used to fit both the GB and the RF
regression models.

Compared to, for example, deep learning methods, GB and RF models can fit properly with relatively small data sets, do not
necessarily require graphical processing units to fit fast, have only a small set of tunable hyperparameters, do not require
heavy preprocessing of the predictor or the target data, such as removal of seasonality. In other words, they are generally
easier to use. That said, one preprocessing step was found to improve modelling accuracy: quantile transformation with $10^5$
quantiles was used to make the target variable, i.e., the $CO_2$ flux, strictly Gaussian distributed. Validation of the model was
then performed using the inverse transformed (non-Gaussian) flux data. The reason for the better results with the Gaussian
transformed data is likely the better modeling of the non-extreme values and the use of the RMSE as a cost function, which
penalizes highly the erroneous extremes: as the great majority of the data is non-extreme, even a slight enhancement of
simulation of the "major bulk" of the data can lead to overall skill improvements – despite the potentially less accurate
simulation of the tails.

Both the GB and the RF are ensemble based tree methods, which means that the final prediction of the model is formed by
calculating the mean of weak learners, trees, constituting the ensemble. Variation between the ensemble members is created
by fitting the members to random subsamples of the predictor matrix X: these subsamples are formed by sampling randomly
both the predictor and the time step dimensions. While the members of the RF are just the trees fitted independently to
different subsamples, the GB takes an additional step by fitting the models hierarchically one by one, such that each member
tree reduces the prediction error of the previous one. In other words, each new member is forced to concentrate on those
observations that are the most difficult to predict correctly (Chapter 10 in Hastie et al., 2009), and in this sense, GB learns
more than the RF.

**2.5 Cross-validation framework, Bayesian parameter tuning, validation metrics**

Repeated K-fold cross-validation with shuffling and R = eight repeats, each divided to K = five folds was used to fit 8×5 = 40 separate ensemble models such that each of the models has its own validation set, and the remaining data was used to fit the model (Hawkins et al., 2003). The validation sets of the five splits together comprise a continuous time series covering all time steps in 1996–2018, and the eight repeats together comprise an ensemble of modeled realizations of the data. We use the ensemble mean over the eight realizations as the best-guess surrogate for the modeled time series.


As an important variation to the standard repeated K-fold cross-validation method, we randomly sampled years instead of individual time steps. Sampling randomly time steps would lead to sampling from the same weather events, i.e., from serially correlated data, which would lead to overestimation of the model accuracy in the validation (Roberts et al., 2017). This can be avoided by sampling sufficiently large, continuous blocks in time, such as years.


For the Bayesian optimization of the hyperparameters, the BayesSearchCV algorithm of the Scikit-Optimize package was used (https://scikit-optimize.github.io/; Snoek et al., 2012). In the gaussian-process based optimization of the algorithm an implicit 5-fold cross-validation with 50 iterations has been used for each of the K = five folds of the dataset. Because of the computational costs, only the first repeat of the cross-validation was used. For the remainder of repeats, R = [2 … 8], the

medians of the optimized parameters were used in fitting. The tuned hyperparameters of the models and their search spaces are listed in Table 2.

For measuring the goodness of fit of the validation samples in the cross-validation, the coefficient of determination ($R^2$-score), the root mean squared error (RMSE), and the Pearson correlation coefficient have been used as metrics of model

skill, and they were calculated from the 6 hourly and weekly data separately.

**Table 2. Optimized hyperparameters of the GB and RF regression models. The root mean squared error was used as the cost function in the Bayesian optimization. Constant default values of other model parameters were used, and they are not presented here. Identified median values of parameter, as well minima and maxima inside brackets, are shown.**

| Model parameter | Explanation | Search space | Optimized value for GB | Optimized value for RF |
|---|---|---|---|---|
| learning_rate | Step size of the optimization process | $\mathbb{R}[0.01 – 0.7]$, log-uniform distribution | 0.030 (0.020 – 0.046) | – |
| max_depth | Maximum depth of a single tree | $\mathbb{Z}[3 – 18]$, uniform distribution | 9 (7 – 18) | 16 (13 – 18) |
| alpha | The L1 regularization parameter | $\mathbb{R}[1\times10^{-9} – 1]$, log-uniform distribution | $1\times10^{-5}$ ($1\times10^{-8}$ – $6\times10^{-3}$) | $1\times10^{-9}$ ($1\times10^{-9}$ – $2\times10^{-9}$) |
| subsample | Random sample size of a tree (proportion of time steps) | $\mathbb{R}[0.01 – 1]$, uniform distribution | 0.73 (0.26 – 0.77) | 1.0 (0.81 – 1.0) |
| colsample_bytree | Random sample size of | $\mathbb{R}[0.01 – 1]$, uniform | 0.10 (0.03 – 0.93) | – |

| | | | | |
|---|---|---|---|---|
| | a tree (proportion of predictors) | distribution | | |
| colsample_bynode | Random sample size of each layer inside a tree (proportion of predictors) | $\mathbb{R}[0.01 - 1]$, uniform distribution | – | 0.10 (0.10 – 0.10) |
| n_estimators | Number of boosting rounds | $\mathbb{Z}[10 - 1000]$, uniform distribution | 730 (240 – 900) | – |
| num_parallel_tree | Number of random forest samples | $\mathbb{Z}[10 - 1000]$, uniform distribution | – | 1000 (640 – 1000) |


## 2.6 SHAP value analysis for measuring the predictor importance

The Tree SHapley Additive exPlanations (Tree SHAP; https://shap.readthedocs.io; Lundberg et al., 2020) is a toolbox for calculating and visualizing the predictor importance. Compared to many other metrics of measuring the importance, such as the gain parameter of the XGBoost, SHAP values are both consistent and accurate, and therefore, more robust (Lundberg, 215 2019). SHAP values were calculated from the validation samples of the models.

## 2.7 Subsampling for artificial reduction of fitting data

Additional subsampling with sample sizes of 10%, 20%, … 100% were used to resample the data within each fold of the cross-validation to give the models less data to learn from. This allows us to measure the sensitivity of the modeling to the 220 amount of fitting data.

The subsampling was implemented with two different strategies. First, the ordinary *random sampling* was used. This strategy mimics cases in which the time series of a site is incomplete, i.e., contains missing observations randomly distributed over the study period. Second, *non-random sampling* was used to study those cases in which the study period is 225 shorter but more complete, implemented by using the same percentage shares as with the random sampling, but selecting continuous blocks of data from the beginning of the cross-validation samples.

## 3 Results

### 3.1 Goodness of fit of the machine learning approaches

For the 6 h GB data, the 95% confidence intervals (CIs), based on bootstrapping with 1000 samples, were 0.910–0.920 for 230 $R^2$, 1.13–1.20 µmol m$^{-2}$ s$^{-1}$ for RMSE, and 0.956–0.961 for correlation (Fig. 3). For the weekly GB data, the 95% CIs were 0.963–0.966, 0.458–0.483 µmol m$^{-2}$ s$^{-1}$, and 0.981–0.983 for $R^2$, RMSE and correlation, respectively. The RF performance was also good, but did not reach the GB skill, as the CIs for the 6 hourly (weekly) data were 0.877–0.891 (0.951–0.955) for

$R^2$, 1.26–1.33 µmol m$^{-2}$ s$^{-1}$ (0.510–0.535 µmol m$^{-2}$ s$^{-1}$) for RMSE, and 0.946–0.952 (0.978–0.980) for correlation, respectively.


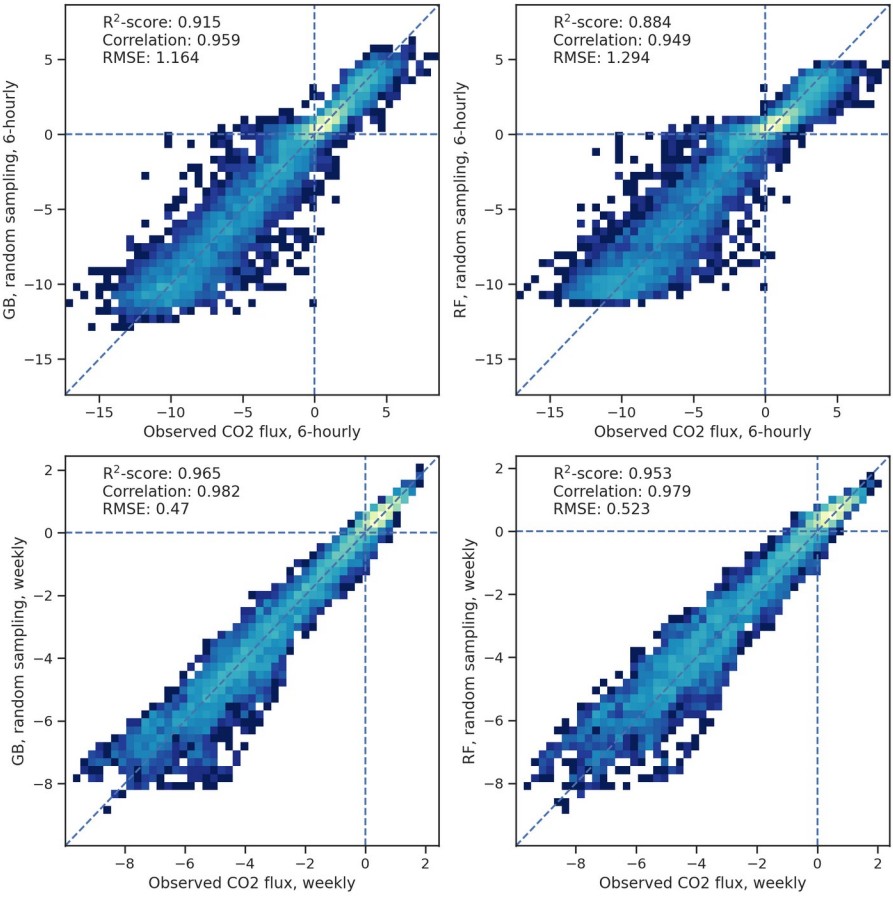

**Figure 3: Two-dimensional probability density histograms of the 6 h (upper row) and weekly mean (bottom) observed and cross-validated modeled CO$_2$ fluxes (GB shown on the left column; RF on right). Color shading indicates qualitatively the density of the observed–modelled value pairs inside each pixel. Bootstrap-estimated medians of R$^2$-scores, Pearson correlation coefficients and the root mean square errors of the fit are also shown. See text for the confidence limits of these values.**


To study the accuracy of modeling without the seasonal and diurnal cycles, monthly and 6 hourly grouping were used simultaneously, and all three quality metrics were calculated for these groups for the GB algorithm (Fig. 4a). This analysis reveals how much the high skill reached in the analysis of the complete time series (Figure 3) is actually attributable to the modeling of the two important temporal cycles. The lowest correlation was found in August at 00 UTC (correlation = 0.43; 245 95% CIs 0.40–0.50) and the highest in April at 06 UTC (0.85; 0.83–0.86). In general, the small absolute values of the flux in

general increase the correlation uncertainty in winter, and on the other hand, the largest variation of the target variable in summer daytime (06–12 UTC; 09–15 local time) yields the largest RMSE, even though the correlation peaks at the same time.

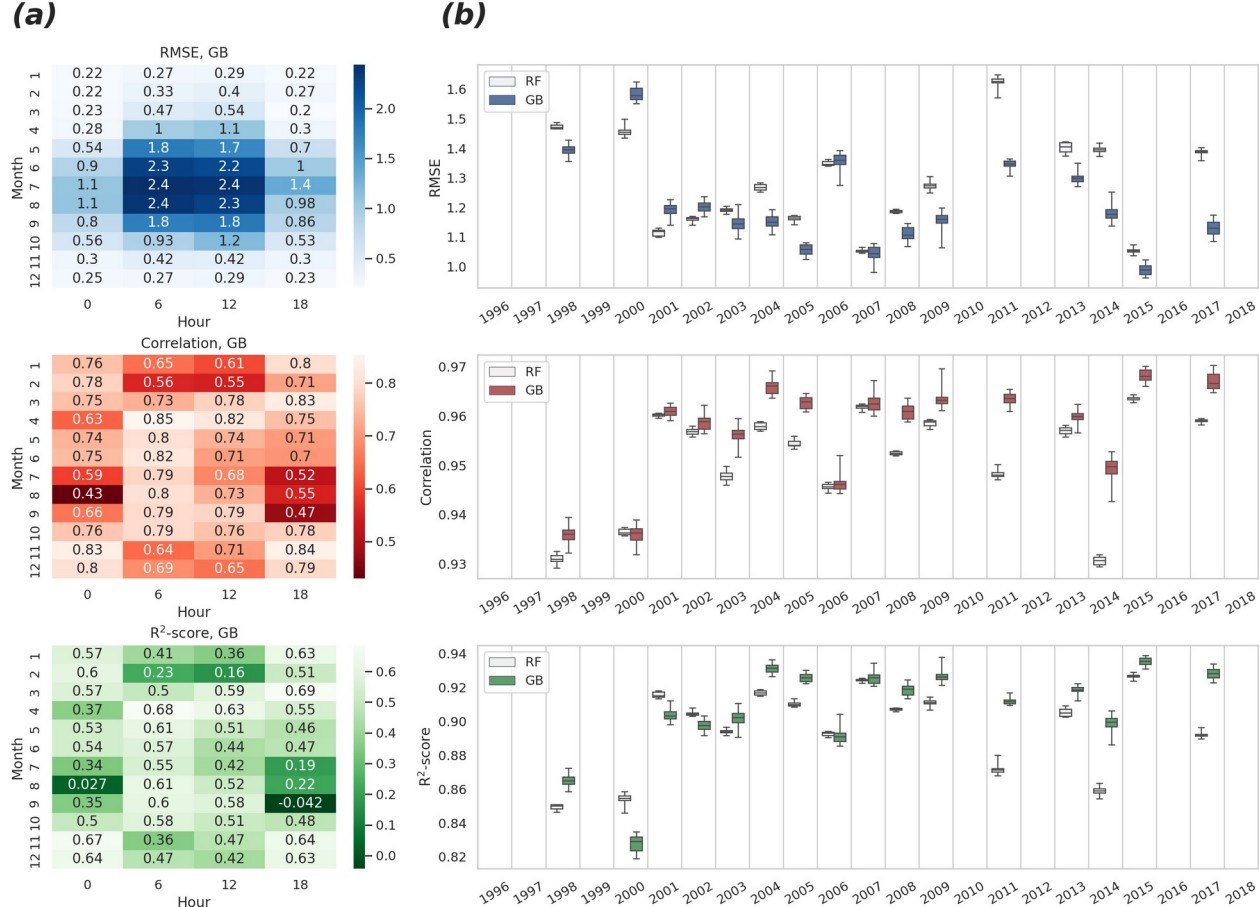

**Figure 4: Estimates of the root mean square error (upper panels), the Pearson correlation coefficient (center panels), and the $R^2$-score (bottom panels) derived from the eight repeated representations of the cross-validated time series. (a): Monthly and 6 hourly decompositions of RMSE, correlation, and $R^2$ for GB. Median values of quantities are shown, calculated from differently repeated cross-validation experiments. (b): Annual variability of RMSE, correlation, and $R^2$ for the RF and GB: variability shown in boxes was calculated from differently repeated cross-validation experiments. Median is shown with a horizontal line in the center of each box. Quartiles are shown with box edges and minima/maxima with whiskers. Years with more than 40% of missing data were excluded.**

When excluding the winter months, the day-time NEE was better predicted ($R^2$ = 0.42–0.68) than the night-time ($R^2$ = -0.04–0.54). Interestingly, an opposite result was achieved when the raw $CO_2$ flux was modeled instead of the preprocessed NEE: in that case the night-time (18–00 UTC) fluxes were better predicted than the morning and afternoon fluxes (not shown).

Analysis of the results of the different target data imply that the sampling error emerging from a rather large share of missing samples in the raw NEE data could explain the differences.

Additionally, annual grouping of the data was used to obtain annual estimates of $R^2$, correlation, and RMSE and their confidence intervals for both ML algorithm results (Fig. 4b). In this case, the CIs were calculated from the distribution of the

different repeats of the K-Fold cross-validation. These estimates show an increasing temporal trend for $R^2$ and correlation, implying either 1) a quality improvement in observed fluxes or 2) in the ERA5 predictor data over the years, or 3) changes in the environment as the forest grows. The highest $R^2$ was achieved in 2015 (median $R^2$ for GB = 0.936), and the lowest in 2000 (median $R^2$ for GB = 0.829).

**3.2   Temporal distribution of the fitting data affects the goodness of fit**

The time series of the $CO_2$ flux observations in Hyytiälä are exceptionally long and complete in time. Therefore, it is interesting to study the sensitivity of modeling to the amount and distribution of fitting data to assess whether the methods could be used for sites with less data. For this, additional subsampling was used to reduce the amount of data prior to fitting of the models in the cross-validation framework.


The results indicate that the GB can cope better with less data compared to RF (Fig. 5). For example, when considering the non-random sampling, the GB achieves the same skill with 20% data as the RF with 70%. Additionally, $R^2$ results reveal that the selection of the ML algorithm is more important in determining the goodness of fit of the result than the selection of the sampling strategy at each percentage level. The differences between random and non-random sampling results also indicate

that lengthening time series by adding more years to it might be a better strategy to further improve the model than gap-filling the missing values in the existing observational time series. This can be seen in the larger changes in the non-random sampling results as the amount of data increases: with the random sampling approach, the changes, and hence the algorithm improvements, become quite small with larger than 60% amounts.

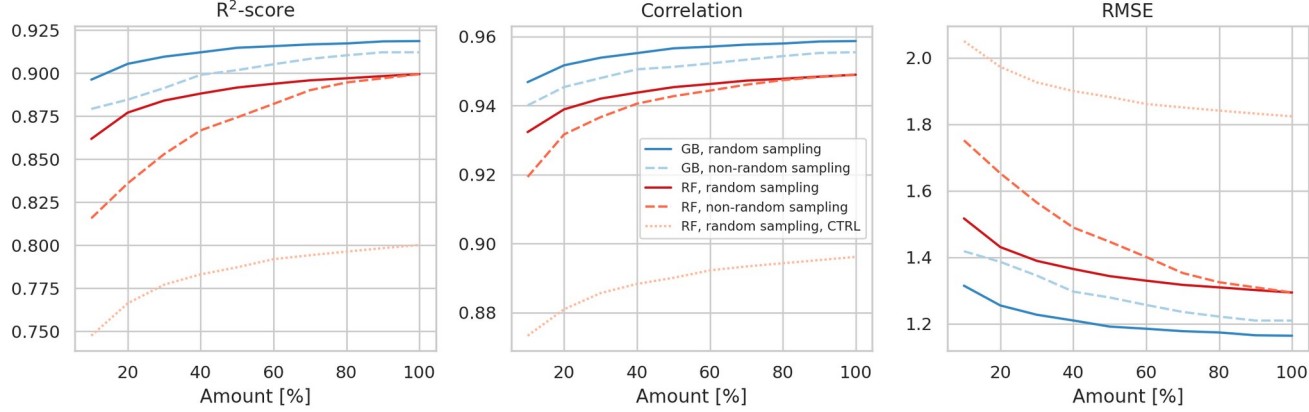

**Figure 5: Cross-validated $R^2$-score (left), the Pearson correlation coefficient (center), and the root mean square error (right) as a function of the amount of fitting data for the gradient boosting approach (blue) and the random forest approach (red). Different subsampling approaches (random versus non-random) are also shown with different dashes. The control experiment – a fit without spatiotemporal neighbors – is shown for the random forest approach. The control for the gradient boosting fails to fit properly with the limited predictors, and it is not shown for this reason. 6 h averages were used in this experiment.**

### 3.3 Analysis of predictor importance

For measuring the predictor importance, SHAP analysis was used. The SHAP implies the relative contribution of each predictor to the model, and it is calculated by measuring each predictor's contribution to each tree of the model. When comparing the predictors, a higher SHAP value implies that the predictor is more important for generating a prediction.

Figure 6 presents the different group means of the predictors in the 40 fitted GB models (which differ from each other by the cross-validation samples used in fitting). The panels a) – c) summarize the mean absolute SHAP results for different parameters, grid cells, and lags. The 2-meter temperature turned out to be the most important of the input parameters. Also, the sensible heat flux, the relative humidity, the short-wave radiation, and the evaporation rate were among the most important predictors. They were followed by the soil temperature of the uppermost layer, the wind components, the latent heat flux, amd the soil temperature of the third layer. The non-lagged variables were more important than the lagged ones, but perhaps surprisingly, the negatively lagged variables turned out to be as important as the positively lagged ones. Contrary to the time dimension, the nearest data point in the spatial dimensions did not contain the most important predictor data on average: the two most important cells, numbers one and five, locate at the bottom corners of the domain (Figure 2b).

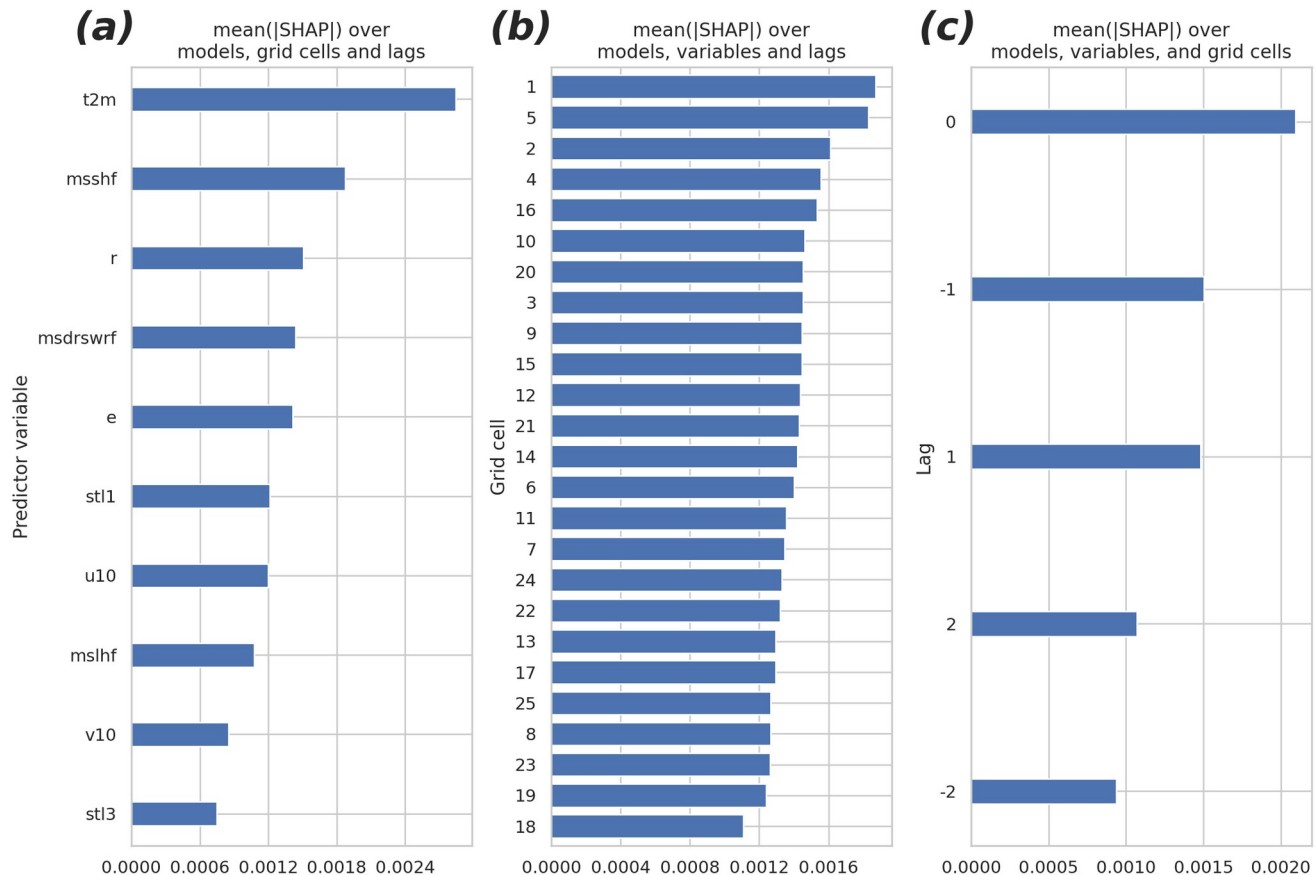

**Figure 6: The mean absolute SHAP values separated by (a) the predictor variable, (b) the grid cell, and (c) the temporal lag. The higher the value, the more important the predictor/cell/lag. The redundant predictors, shown in Table 1, were excluded from the analysis and from the final fit of the model. See Figure 2 for the organization of the grid cells and temporal lags.**

To study the overall relevance of the input variables, we conducted an experiment in which we excluded them one by one, beginning from the one with least explanatory power (total cloud cover), and measured the accuracy of GB after each drop until it started to decrease significantly. It turned out that half of the variables originally included were redundant, i.e., they did not improve the accuracy at all. Importantly, however, they did not worsen the models significantly either. Soil moisture of the third layer was the first variable to add significant value to the models, and those with a smaller average gain could be discarded without major effects to the results. Among the all input variables, the top-six – those placing above the 10-meter u-wind component in Figure 6a – were the ones to improve the model the most. Interestingly, using only the two most important variables, the 2-meter temperature and the sensible heat flux, yielded a model with a relatively good accuracy ($R^2$ = 0.87), corresponding the best accuracy achieved with the RF model with all available predictors.

## 4 Discussion

Many local factors affecting net $CO_2$ exchange between the atmosphere and a boreal forest either vary only slowly over time, as is the case for the plant distribution and growth and soil microorganism populations, or are effectively constant (e.g., soil properties and shape of the terrain). In contrast, the variability of meteorological factors is prominent and happens in short time scales and, partly for these reasons, dominates the variability of the flux response (Sierra et al., 2009). Indeed, the vast majority of the $CO_2$ flux variation in the studied forest can be explained by using only meteorological factors, of which the most important ones were, in order, air temperature, sensible heat flux, relative humidity, short-wave radiation, evaporation rate, soil temperature, wind components, and latent heat flux. Out of all 19 variables included in the analysis, these are the ones which significantly contributed to the GB model skill. It is worth noting that some of the variables included in the analysis are not completely independent of the physical and biophysical processes: to some extent, many of them are regulated by the plants themselves, and the environment in general. The most important of such variables are the latent and sensible heat fluxes, evaporation, relative humidity, and the near-surface temperature.

At least to some extent, if not completely, the ML methods employed here might be able to account for slow changes in the response happening over the years if 1) they are caused by the meteorological variables, and 2) the current period of the study contains clear enough signals of these changes. For example, the increasing trend in temperature is one of the most important variables explaining the $CO_2$ variability both in the short and long term (Huntingford et al., 2017; Pulliainen et al., 2017). However, the presented methods can not extrapolate cases in which the values of a predictor variable fall outside of the range used in fitting the models. It is likely that the temperature extremes exceed the observed variability in the near future along with the warming local and global climate. The sensitivity of the predicted NEE on the temperatures residing outside of the observed range remains unclear, but eventually, the ecosystem changes become so large that the accuracy of the method will necessarily deteriorate.

When interpreting the results, it is important to distinguish the conceptual difference between the negative and positive temporal lags. A strong correlation between the response variable and positively lagged predictor is an indicator of the predictor driving the $CO_2$ flux, either directly or indirectly. A correlation between the flux and a negatively lagged predictor variable is more difficult to understand. It is likely that because of temporal biases and other inaccuracies in the gridded form of the variables, some of the negatively lagged predictors might better represent the relevant variability for modeling. Similarly, because of spatial biases, some of the neighboring grid cells might better represent the local conditions than the nearest cell: in our experiments, the most useful predictor variability was found in the bottom corner cells of the domain.

In general, machine learning methods seek for relationships between the response variable and the predictor data, and they cannot distinguish whether these relationships are truly causal. Even though the identified relationships and interaction

mechanisms may not be intuitive and even causally coherent, they can still be used to improve the model accuracy. To be beneficial for the modeling, such a relationship just needs to be sufficiently robust and strong, and constant in time. Even though the predictor dataset contained many redundant variables, the GB method gave them a low enough feature importance, and hence, the cross-validated correlation remained high. The effectiveness of the GB in rejecting the irrelevant predictors and variability was also evident in the pre-processing: the principal component analysis, which often helps ML

models to find the most important dimensions of the preidctor data, did not improve the skill at all. In addition to this, the method proved to be skilful even in cases in which the amount of fitting samples was heavily reduced. With less powerful statistical methods, overfitting would be much more likely, leading to poorer cross-validation results when using redundant and/or collinear predictor variables and/or small fitting samples (Chapter 7 in Wilks, 2011; Chapters 3 and 7 in Hastie et al., 2009; Lavery et al., 2019).

Both the efficiency of the GB method in omitting the non-optimal predictors and the ability to cope with small fitting samples are especially encouraging considering its application to other locations: all variables can be used, letting the model decide about the redundancy. It is likely that the same variables that were found important at our study site might not constitute an optimal choice in other ecosystems and locations; vice versa, the predictors found redundant in Hyytiälä, such

as soil moisture, can be important in other environments (Nadal-Sala et al., 2021; Zhou et al., 2019).

This work could act as a first step in creation of a multi-purpose, national, regional, or global flux model (Jung et al., 2020; Bodesheim et al., 2018), because 1) the meteorological predictors can explain almost all of the variability of the observed atmosphere-ecosystem NEE, 2) GB regression is efficient in modeling that variability, 3) NEE is measured globally at a

375 large number of sites representing different climates and ecosystems (Hicks and Baldocchi, 2020), and 4) the meteorological variables, derived here from the ERA5 reanalysis, are easily and freely available globally in a spatially and temporally dense, complete, and homogeneous format, extending back to the 1950s. However, for that, a transformation of the model from modeling only one dimension (time) to modeling of three dimensions (time, latitude, longitude) would be necessary, requiring an abundant set of NEE observation samples representing different bioclimates, and additionally, spatial

information about the biology and geography (vegetation, land properties, orography, latitude, etc.) of those locations would be needed to allow the model learn the spatiotemporal relationships between the predictor variables and NEE. It is also worth noting that spatiotemporal structures that the models learn and utilize from the predictor neighborhoods of the meteorological data might not easily and directly translate to different locations.

Another, more easily attainable application for the proposed spatiotemporal approach is to use it for gap-filling of the EC measurements of NEE, which are typically available in the half-hourly resolution (Pastorello et al., 2020). In that context the spatiotemporal structures can be directly learned for each of the study sites separately. For gap-filling, the ERA5 data should

be downloaded in the full 1 hourly resolution and resampled to half-hourly resolution. The applicability of the method in that time resolution remains to be tested, but as shown with the current experiments, it works well for gap-filling the NEE data in the 6 hourly time resolution. Compared to other studies of this kind, our results are promising in terms of the $R^2$ skill (Irvin et al., 2021; Mahabbati et al., 2021).

## 5 conclusions

As summarized in Figure 5, the combination of novelties of this study, namely using GB, which excels in this context compared to RF, and using the spatiotemporal neighborhoods from the meteorological input data together yielded a high level of accuracy in modelling both the subdaily and weekly variability of the atmosphere–forest $CO_2$ exchange. Even though the time series of our study were exceptionally long, the GB could cope with much shorter time series as well. As such, the approach is almost directly applicable to gap-filling of the observational NEE data. However, for application of the method in the multi-site context, new stationary predictors would be needed, and the accuracy of the model should be measured using, for example, a leave-one-site-out cross-validation strategy (Roberts et al., 2017).

## Code and data availability

The code for reproducing the results from experiments and analyses is available at Kämäräinen et al. (2022; https://zenodo.org/badge/latestdoi/368864113). The code can be used to download and preprocess also the ERA5 predictor data: other data, including NEE data, are included in the repository.

## Author contribution

MKä designed the experiments and the structure and content of the manuscript, wrote and executed the code, and composed the text. ALi participated in the planning of the manuscript content and made major suggestions during the writing process, and helped significantly with the references. JT contributed significantly to the content of the reference list and commented the text. IM was responsible for the EC measurements at the study site. HV tested the code and made suggestions how to improve it. MKu, JA, and ALo commented the manuscript.

## Competing interests

Authors declare that there are no competing or conflicting interests affecting the work.

**Acknowledgements and financial support**

We thank two anonymous reviewers for their valuable comments and suggestions to improve the manuscript. We thank the
creators and maintainers of the ERA5 reanalysis for providing this invaluable data freely available for the research
community. We also thank Hyytiälä SMEAR II staff, ICOS research infrastructure and the responsible researchers for
maintaining the eddy covariance data and providing it openly available online. We acknowledge the following projects for
the funding of the work: ACCC Flagship funded by the Academy of Finland (337549); Academy professorship funded by
the Academy of Finland (302958); research projects funded by the Academy of Finland (342890, 325656, 316114, 325647,
347782); Jane and Aatos Erkko Foundation (project Quantifying carbon sink, CarbonSink+ and their interaction with air
quality); the European Research Council project ATM-GTP (742206); the project "Mitigation and adaptation of carbon
sequestration by co-creation" (HIILIPOLKU), funded by the Ministry of Agriculture and Forestry in Finland, grant no.
VN/28443/2021-MMM-2 (Catch the Carbon—program).

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
