# Peer review of "Evaluation of gradient boosting and random forest methods to model subdaily variability of the atmosphere–forest CO2 exchange"

_Biogeosciences, 2022_

## Author Response (AR1)

Biogeosciences Discuss., referee comment RC1
https://doi.org/10.5194/bg-2022-108-RC1, 2022

[Figure]

**Comment on bg-2022-108**

Anonymous Referee #1

Referee comment on "Evaluation of gradient boosting and random forest methods to model subdaily variability of the atmosphere–forest CO2 exchange" by Matti Kämäräinen et al., Biogeosciences Discuss., https://doi.org/10.5194/bg-2022-108-RC1, 2022

This study by Kämäräinen et al. compares two different machine learning algorithms for the prediction of $CO_2$ net ecosystem exchange of a boreal forest using ERA5 reanalysis data. Getting accurate estimates of $CO_2$ exchange outside of spatiotemporal domains covered by eddy covariance measurements is an important task and hence a relevant topic for Biogeosciences. The analysis of spatial and temporal neighborhood predictors and the emulation of less complete time series are interesting concepts and the study is overall well-written and structured. However, I have some general and specific concerns and questions listed below that should be addressed in a round of major revisions before publication.

**General comments:**

Introduction/Discussion: I think a more complete consideration of state-of-the-art literature could better present the novelty of this study which is not clear in the current form, and that literature should also be considered more for discussing the results. For example, please show what improvements can be expected from gradient boosting in view of previous research, e.g. Tramontana et al. 2016 (https://doi.org/10.5194/bg-13-4291-2016), who already compared various ML algorithms for NEE and GPP prediction. Please also cite literature regarding spatiotemporal neighborhood predictors, if there is any.

We thank the reviewer for the overall positive comments on our manuscript. We agree that both the Introduction and Discussion chapters could include more references to the previously published articles, and we are planning to include them in the corrected manuscript. However, as the time resolution and other details are quite different between different studies, the direct comparisons of the results between different studies might not be possible.

References to temporal lagging of the predictor data (temporal neighborhood) should be quite easy to find and we will include those: however, the utilization of the spatial neighborhood is perhaps a new invention and references for that approach are much harder to find for this reason.

In the Introduction we added Bodesheim et al., 2018; Snoek et al., 2012; Besnard et al., 2019; Lundberg et al., 2020; Tramontana et al., 2016. We extended the Introduction by adding more discussion about the previous work. We could not find any references for the spatiotemporal neighborhood approach.

We also extended the Discussion section, for example by including discussions about the potential of the proposed approach as a gap-filling method.

As this study seems to be conducted in view of the general goal of estimating carbon fluxes for points in time and especially in space without direct flux observations, it would be more interesting to test the models also with spatially independent data, i.e., for a comparable boreal EC station (from Fluxnet for example) fully excluded from model training. Otherwise, it should be pointed out more clearly that the predictive error likely is much higher when the models are applied to new locations, see e.g., Roberts et al., 2017 https://doi.org/10.1111/ecog.02881

Our approach models the NEE only in time dimension: it does not receive any spatial information, and for this reason it can not be generalized outside the study site. To really make it applicable to other locations, a transformation of the model to three dimensions (time, latitude, longitude) would be necessary. For this an abundant set of NEE observation samples representing different (boreal) bioclimates would be necessary, and additionally, spatial information about the biology and geography (vegetation, land properties, orography, latitude, etc.) of those locations would be needed to allow the model learn the spatiotemporal relationships between the predictor variables and the NEE.

Applying the model as it is in different sites would implicitly contain an assumption that, for example, the vegetation and soil properties are the same everywhere, which of course is not a realistic assumption.

We will make this more clear in the text.

This was made more clear and explicit in the Discussion chapter. Also, a new Conclusions chapter was added, and it was also pointed out there.

While I see that ML-models do not require causal relations, I'm still not convinced by the inclusion of negatively lagged, i.e. future, meteo-variables. Did the exclusion of negatively lagged variables actually deteriorate model accuracy or are they just redundant with spurious correlations? In any case, the explanation regarding advection requires references as a theoretical basis supporting it. To me, it makes sense only for grid cells downwind from cells representing the station well (e.g. the central cell). However, I still don't see what additional information can be gained as all the "advected" information already is contained in the non-time-lagged data of the more representative grid cells. Furthermore, it makes no sense for all meteo-variables, e.g., radiation and soil temperature, two of the most important variables, certainly are not advected directly. Hence, I think this explanation requires a more profound basis, i.e., by analyzing the importance of negatively lagged variables by grid cell in relation to wind direction, and by meteo variable. Otherwise, negatively lagged variables should rather be excluded from the analysis in my opinion.

The reason why they have been included in the first version of the manuscript are the spatiotemporal uncertainties of the ERA5 data. The reanalysis is a **modeled representation** of the observations of the meteorological variables, and for this reason it necessarily contains uncertainty, such as biases. Letting the gradient boosting learn the spatiotemporal neighborhood of the data makes the model able to actually learn – and take into account – the effect of the spatiotemporal biases.

We will consider excluding the negatively lagged time steps from the predictor data: that will not deteriorate the results too much, but as they seem to raise questions and confuse readers, it might be better to not use them. If we use the negative lags also in the next version of the manuscript, we will make sure the reasons why they were regarded as useful.

The new experiments show that the negatively lagged predictors contain very useful information, and they were included in the corrected manuscript for that reason. The text was extended and reworded to explain better the potential reasons for their usefulness: see Section 2.3 and Discussion.

The Pearson correlation coefficient is insensitive to magnitude, so it does not tell how accurate the predicted values are and hence is not very meaningful for model evaluations. I recommend to focus on R2 instead.

We agree with Referee #1 that the Pearson CC does not take into account and measure the variance or bias of the data. However, using R2 instead of Pearson CC when evaluating the goodness-of-fit would not likely change the results, because the R2 measures the linear correlation quite similarly as the Pearson CC a) when bias is very small, b) when skill is high, and c) when the metrics are evaluated from large samples. All conditions, a), b), and c), are fulfilled in our modeling.

https://en.wikipedia.org/wiki/Coefficient_of_determination

See Figure 1 for comparisons of Pearson CC and R2 in a set of random simulations of correlated datasets. See also our response to general comments of Referee #2.

[Figure]

*Figure 1. Relation of the square of the Pearson correlation coefficient and the R2 score in unbiased random (but correlated) datasets. Each point represents one dataset of size (250,2): the Pearson correlation and R2 score were calculated between the two columns.*

*Using larger sample sizes than 250 (our CO2 data contains 10000 samples) would make the relationship even stronger.*

Because Pearson CC does not measure the variance or bias, we have used the RMS as an alternative measure of the skill, as RMS is affected by the variance and bias of the modeled data. When we draw conclusions about the goodness-of-fit, we take into account both measures: the Pearson CC, and the RMS.

We can change the analysis such that R2 (or NSE as suggested by Referee #2) will be used instead of the Pearson CC, as it better takes into account the bias and variance in a

single metric, but very likely it will not have major effects on the conclusions.

We included the R2 score in the analysis.

**Specific comments:**
L24-25: This recommendation is too general, as the models have been evaluated just for one specific ecosystem.

We will modify this recommendation such that it better takes into account the limitations of the data.

The sentence was reworded to also allow opposite cases.

L45-48: Please add a reference here.

We will consider adding a reference. However, this piece of general knowledge might not necessarily need one in our opinion.

A reference was not added.

L65: Is the reference to kaggle really necessary? This rather comes across as an advertisement for a company, so please remove it.

We will remove the reference.

The reference was removed.

L80-81: Please make clear that EddyUH (and REddyProc?) processing was not done within this study but before data was acquired.

We will modify the sentence, for example like this: "Flux processing for the NEE was done previously by Mammarella et al. (2016) using the EddyUH software (a summary of the data is shown in Fig. 1, presented as multi-year mean values)."

The sentence was modified.

L81-83: Rather explain NEE when the term is first introduced in Section 1.

We agree that this sentence would be better to locate earlier in the text: we will move it to Section 1.

NEE is now explained in Section 1.

L85: What constitutes a missing value? Were flux data filtered according to a certain quality control strategy, e.g. a test on stationarity, well-developed turbulence, footprint etc.?

Because the flux processing was done earlier, as mentioned by Referee #1, we do not take a closer look into the constitution of the missing data in this article. In general, the very raw data contains missing values due to technical faults in the instruments,

power outages, and so on. Additionally, the flux processing is an additional filter, which causes some other data to be discarded, as suggested by the Referee. And finally, the averaging process discards a major part of the data as explained in the text.

We can extend the sentence to make it more clear, for example: "...windows. Only complete 6 hourly aggregates, i.e. those with no missing values arising from flux processing and instrument faults, were accepted for the averaging process."

The sentence was extended.

L90 (Fig. 1): Why is 1998 written below Jan? The title seems superfluous, rather add NEE to the y-axis.

The year 1998 was accidentally left in the figure: it will be removed in the next version of the draft.

We will shorten the title and add NEE to the y-axis.

The year was removed.

L95-96: Were missing values gap-filled or just omitted for the calculation of the weekly means? The latter would likely introduce a bias towards more negative NEE values as likely more nighttime data are missing compared to daytime data. This could at least be mentioned.

The missing values were omitted from the calculation of the weekly means. However, the same missing steps were also removed from the modeled data: this makes quality-of-fit measures fair.

We can mention the diurnal distribution of the missing data in the text, even though it won't affect the quality-of-fit/skill estimation of the model.

The diurnal distribution of the missing data is shown in the text.

L107: Are you sure 1° is the spatial resolution? I think it's 0.1°, otherwise it would be really coarse.

The 1° resolution data is what we have been using here. The original data is 0.25°, which is of course denser. We can download the data in this denser resolution and re-calculate the results, and if the new results are significantly different, we will change the text and figures accordingly.

The data was downloaded in the denser resolution, and new results are now reported accordingly. The overall impact of using the denser resolution on the accuracy of modeling was slightly positive but not likely significant.

L110: Some of the abbreviations appear quite bulky, rather use more common ones like H, LE and rH. Also, is diffuse or total shortwave radiation not available in the ERA5 product?

These abbreviations were inherited from the ERA5 data. We are not completely sure whether more common abbreviations actually exist for all quantities which all readers

could accept, but we will consider whether we can improve the readability of the presentation of the variables.

Different shortwave variables are available in the ERA5. We will consider if adding one or more of them in the list of predictors would offer significant added value for the study.

We decided to follow the ERA5 naming of the variables. We also tested the total SW flux, but it did not enhance the accuracy of modeling compared to using the direct SW flux.

L123-127 (Fig. 2): in the caption, temporal lags from -2 to +2 are stated, though in the figure lags from +3 to -1 are visualized. Please also make the numbering uniform between Fig. 2 and A2, i.e. that the central cell is number 13 in both figures and so on. As some of the most important grid cells are outside the inner circle (10, 11 ,21), I think it's necessary and less confusing to show them all.

We will modify the figure according to the suggestions of the Referee.

The figure was modified as requested.

L129-135: Were all 23752 operations carried out in the end (as PCA was not used) despite technically being too laborious? Please clarify.

Unfortunately, we could not carry it out, as the computation would have taken too much time.

L165-168: Please add a reference.

We will add a reference.

A reference was added.

L171-172: How many data points were included in each of the $10^3$ bootstrap samples?

We have used standard bootstrapping. In it the same number of data points is sampled as in the original data: 10500. Variation between the samples is caused by sampling **with replacement**. See https://en.wikipedia.org/wiki/Bootstrapping_(statistics)

L201-202: Are 00 and 06 UTC the start or the end of the averaging period? (also relevant for Fig. 4)

It is the beginning of the period. We will mention this in the corrected text.

This is now mentioned in Section 2.1.

L242: Rather write "cope better" as this is not a yes-no question. To evaluate which one copes better, wouldn't it also be necessary to compare the decrease in prediction skill of each model to its own 100% CORR and RMSE values?

We will add the word "better" to that sentence.

We are not quite sure what Referee #1 means with the latter comment. At 100% there is no difference between the sampling procedures – the data are literally the same. This is also visible in Figure 5: the lines of the same color (dashing for non-random and solid line for random sampling) merge after 90%, and at 100%, the values are the same.

The word "better" was added.

The experiment was also redesigned to use the Repeated KFold CV instead of the standard KFold, which adds robustness to the results.

L258: Are highly correlated variables an issue for gain? I know they induce a bias for permutation importance, so can this be excluded for gain?

This is an interesting question, but we do not have a direct answer for it.

If two highly correlated variables are present in the predictor data, both will get high gain values if they are relevant for the prediction, but only if random forest style subsampling is used when building the trees. Without subsampling, probably only the better one of those two would get a high gain score (as it would be selected for all trees), the other getting near zero gain (as it would be probably rejected from most trees).

In most cases, and especially with uncertain and noisy atmospheric data, using subsampling is recommended, not only as it yields more accurate models, but also because it likely makes the gain values more stable.

We have used subsampling both for the GB and RF models in this study, which makes their results comparable (even though we did not present the gain from the RF models).

Whether the differences in gain results between the different approaches (subsampling vs. no subsampling) is a problem depends on the desired outcome. Subsampling "softens" the differences between gain values of predictors, which can reveal potentially interesting results.

See also the answers to the question of SHAP values by Referee #2.

We have replaced gain analysis with SHAP analysis, which should be more robust. One consequence was that the order of the variables in the importance listing was changed to some degree.

L266: direct or total SW radiation?

We had only one SW radiation parameter in this study: "Mean surface direct short-wave radiation flux", see Table 1.

L269 (& L312-313): I think this could be worth some more detailed analysis. To what percentage was (pine) forest the dominant land cover in each grid cell? Does this correlate with the importance statistics? Is there any spatial pattern? (Figure A2 could be visualized as a map for this).

Figure 2 shows the typical, scattered landscape of the nearest grid cell of Hyytiälä. The surrounding cells are quite similar. Mostly forest covered, lots of lakes, and when zoomed closer, agricultural land is quite dominant as well. Unfortunately, telling the requested percentages, or examining why this area was not giving the highest gain value in the analysis, might be too laborious tasks.

However, recalculating the results in the denser spatial resolution might change the result – we will see.

The result was slightly different when using the denser resolution data and a different importance metric (SHAP). Now the most important cells are in the bottom corners of the domain. Note, though, that the differences are not that significant anymore (Figure 6b in the manuscript). We believe that the spatio(temporal) uncertainties of the reanalysis explain this result.

The new result is now reported in the text, and reasons explained in Discussions.

[Figure]

Figure 2. A satellite image of the area corresponding roughly to the location of the nearest ERA5 grid cell of the study site.

L275 (Fig. 6): The figure would be more readable if the importance results were averaged over the five models with a measure of variation. Alternatively, swap the figure with one or all figures of the Appendix, as they are more easily recognizable and hence more valuable to the reader. Also, an x-axis label is missing.

We will swap Figure 6 and A1.

The figures from the Appendix were swapped to text, and the Appendix was removed.

L276-279: the same results for grid cells and time lags would also be interesting.

We will consider, but not promise, testing this experiment for the requested dimensions as well.
We could not perform the requested experiments, as they were too laborious, given the resources we had.

L302: Are there any papers investigating the model accuracy for out-of-range data?

We can try to find examples. However, in general, tree-based methods do not extrapolate well outside the range.

We could not find any climate related studies of this. However, the result will be strongly dependent on the number (and extremity) of the cases residing outside of the range.

For a common explanation of the extrapolation issue, see this blog post: https://towardsdatascience.com/xgboost-for-time-series-youre-gonna-need-a-bigger-boat-9d329efa6814

L309-310: "more of a proxy-like" sounds clumsy. Suggestion: "represented by proxy".

We will change the text as suggested.

The sentence was completely removed/reworded.

L339: I think the quoted FLUXCOM approach by Jung et al. already is a global flux model for this very purpose. Hence, the formulation "could act as a first step" sounds rather misleading to me.

Please note that in that sentence we do not refer to creation of **the first** global flux model: we refer to creation of **a** global flux model, and to make one, it has to be started from the first step.

**Technical comments:**
Articles are sometimes used excessively for generic nouns, e.g. L12-16, L.78, L.242-243
Write $CO_2$ with a subscript 2

We will remove articles from generic nouns and formulate "CO2" with a subscript.

Excessive articles were removed from the text.

Biogeosciences Discuss., referee comment RC2
https://doi.org/10.5194/bg-2022-108-RC2, 2022

[Figure]

**Comment on bg-2022-108**

Anonymous Referee #2

Referee comment on "Evaluation of gradient boosting and random forest methods to model subdaily variability of the atmosphere–forest CO2 exchange" by Matti Kämäräinen et al., Biogeosciences Discuss., https://doi.org/10.5194/bg-2022-108-RC2, 2022

This paper (GCB-21-2684) evaluated the predictive skill of two machine learning models for estimating sub-daily net ecosystem exchange (NEE) in a long-term boreal forest site. Although using machine learning to model NEE is not a new topic, this study provides informative results on model choice (XGBoost vs commonly used random forest), the use of climatic data solely to estimate NEE, and the benefits of incorporating spatial and temporal autocorrelated information. These results are potentially helpful to carbon flux modeling with machine learning. I have several outstanding questions and suggestions that I hope the authors would consider.

**Major comments:**

1. The introduction should provide more background on the use of machine learning to model eddy covariance measured NEE and identify the knowledge gap that this paper tries to fill. Many studies have employed machine learning models to upscale eddy covariance NEE, and global products such as FLUXCOM are available. Therefore, what makes this study significant or informative when it models NEE with machine learning in a single site? This paper looks at novel aspects which were not discussed in the introduction, such as comparing GB vs. RF; incorporating spatial and temporal information.

It is true that spatially more comprehensive prior work exists. As mentioned by Referee #2, we have introduced some new ideas on how to model NEE perhaps better, or at least differently, than what has been achieved earlier.

For example, dozens of general circulation models (GCMs) worldwide contribute to the IPCC assessments of the ongoing global climate change. Each of those GCMs model the same common goal – the spatiotemporal variability of the key climate variables – using more or less different approaches. Together their results complete each other. Similarly, different impact models (such as the one presented in our work) could be used to 1) find new ways to achieve the common goal of modeling accurately NEE, to 2) complete the estimations of the (spatio-) temporal variability of NEE, to 3) help other modelers perhaps improve their own approaches, and so on.

We will improve the Introduction by better discussing the novelties of this study, as suggested by Referee #2.

The Introduction chapter was extended and improved.

2. A more rigorous model evaluation procedure would help improve the robustness of the model comparison results. This could include 1) using different types of goodness-of-fit metrics (e.g. NSE and bias), 2) estimating uncertainties of model performance from repeated cross-validation with random splitting and model initialization. Please see my

specific comments.

We can improve the evaluation by including NSE (or R2 as suggested by Referee #1). However, at this point, we do not expect the conclusions to be different, as NSE, Pearson CC and R2 all produce comparable results a) when bias is small, b) when skill is high, and c) when the metrics are evaluated from large samples. All conditions, a), b), and c), are fulfilled in our modeling.

https://en.wikipedia.org/wiki/Nash%E2%80%93Sutcliffe_model_efficiency_coefficient

See Figure 1 for comparisons of Pearson CC and NSE in a set of random simulations of correlated datasets. See also our response to general comments of Referee #1.

[Figure]

*Figure 1. Relation of the square of the Pearson correlation coefficient and the NSE in unbiased random (but correlated) datasets. Each point represents one dataset of size (250,2): the Pearson correlation and NSE were calculated between the two columns.*

*Using larger sample sizes than 250 (our CO2 data contains 10000 samples) would make the relationship even stronger.*

We can change the analysis such that NSE (or R2 as suggested by Referee #1) will be used instead of the Pearson CC, as it better takes into account the bias and variance in a single metric, but very likely it will not have major effects on the conclusions.

See specific comments later for the question about cross-validation experiments.

We included the R2 score in the analysis.

3. It would be interesting to look at how incorporating neighboring temporal and spatial information affects the predictability of NEE by the machine learning models since previous studies usually only focus on concurrent and collocated measurements/inputs.

While the feature importance analysis shed light on the benefits of spatiotemporal information, the importance metrics are difficult to interpret for tree-based models, given that many features are highly correlated. A direct comparison between models with and without spatial/temporally neighboring information would be appreciated.

We can perform a control simulation without the spatiotemporal neighboring data.

Depending on the results we will decide how to report them – either visually in the Figures or in the text only.

Control simulations without the spatiotemporal neighbor information were performed and reported in the text: see e.g. Section 3.2.

4. Global feature importance metrics are sometimes unstable and difficult to interpret for tree ensemble methods, especially when features are highly correlated. I suggest evaluating feature importance using SHAP as an additional metric to get a more rigorous quantification of importance. See some discussions about feature importance here (Yasodhara et al., 2021, https://link.springer.com/chapter/10.1007/978-3-030-84060-0_19#Sec), here (https://towardsdatascience.com/interpretable-machine-learning-with-xgboost-9ec80d148d27), and an example using SHAP here (Green et al., 2022, https://onlinelibrary.wiley.com/doi/abs/10.1111/gcb.16139).

We thank Referee #2 for the references and for the idea of using SHAP values as a complementary or alternative measure of predictor importance. However, we are not sure whether we have enough resources to complete the study with SHAP values. We agree that sometimes interpretation of the gain values can be difficult, but we believe that the subsampling procedure that we used makes the gain analysis more stable: see our response to Referee #1 (the comment about gain values, referring to line 258 in the text).

We will consider, but do not promise, either testing only shortly or also reporting SHAP values in the article. If we can not use them, we will then warn the readers about the potential stability problems of the gain values.

We have replaced gain analysis with SHAP analysis, which should be more robust. One consequence was that the order of the variables in the importance listing was changed to some degree. However, the input data was also changed (denser spatial resolution) and models were retrained using a better hyperparameter optimization strategy. It is thus difficult to say how much the different importance result is attributable to the change of the importance measurement methodology.

5. Data-driven models of carbon fluxes often use satellite observed structural vegetation information as a major input. Therefore, it is interesting to see in this paper, that climate variables (from ERA5) could explain 95% of temporal dynamics of NEE in a site. Moreover, the level of accuracy from this paper is considerably higher than those from similar studies, both from a single site and from spatial upscaling over multiple sites. Could you please provide more discussion on the model performance and feature selection of this study in the context of previous results from the literature?

We will provide more discussion about the accuracy of the results.

The most important reason explaining the good result is the direct availability of the observational data in the study site, which allows the model to learn the site-specific temporal distribution and other details accurately.

Building a full 3-dimensional model (with dimensions (time, latitude, longitude)) would have required more measurements of NEE from sites representing different bioclimatic conditions. That kind of modeling would enable eg. LOO cross-validation over different study sites, yielding estimates of spatial uncertainty, which are not possible to get using

only one measurement site.

Comparisons of our results with single site studies are occasionally difficult because of different time resolutions, different skill metrics, and different bioclimates, but we will try it as well as possible.

See also our response to the second general comment of Referee #1 about details required to make a global or regional model of spatiotemporal variability of NEE.

We extended the manuscript by discussing the difference between our single-site approach and upscaling approaches (see, e.g., 4 Discussion and 5 Conclusions).

**Specific comments:**
**Abstract**
L18-19: This is an informative finding. But the manuscript doesn't have an experiment that directly compares a model with spatial and temporal information to a model without such features.

We can perform such an experiment and report the results in some appropriate way.

We now present the results from the proposed experiment.

L20-22: Both GB and RF rely on the same theoretical approach to identify features that are important to minimize the loss function since they are both tree-based algorithms. The fact that GB is more accurate than RF demonstrates the effectiveness of the "boosting" technique, but there is no direct evidence that GB identifies "more important features" than RF or is more resistant to overfitting.

We can repeat our experiment about inclusion of input variables one by one also for the RF to see whether direct signs of overfitting from that approach could be found.

We did not have time and other resources to perform the same experiment for RF, but instead, we changed the text of the abstract to be more precise.

**Introduction**
L50-56: Background on the reanalysis is informative, but is this necessary for this paper, given that most readers may already have a general knowledge.

Among the authors of the manuscript it was considered important. The educational background of readers of Biogeosciences might not be very homogeneous, and therefore, we think these lines might be good to have there. However, we can also remove the explanation, if requested by the Referee.

Some of the authors considered this piece of text important; we let it be as it was.

**Methods**
L134-135: Does this result apply to both RF and GB? This is an interesting finding to me and could be highlighted in the result/discussion/conclusion.

We will highlight this result in other parts of the text as well.

In addition to Section 2.3, we mention this result now also in the Discussion chapter.

L145: Could you please elaborate on the benefits of transforming the target variable to Gaussian?

Even though it is not completely clear to us why the results are better with transformed (and back-transformed after modeling) data, it might be related to better simulation of the non-extreme values. See this example and the related discussion: https://stats.stackexchange.com/questions/447863/log-transforming-target-var-for-training-a-random-forest-regressor

As the great majority of the data is non-extreme, even a slight enhancement of simulation of the "major bulk" of the data can lead to overall skill improvements – despite the slightly less accurate simulation of the tails.

We will add this explanation in the text.

We now explain the reason for better skill of modeling when Gaussian transformation is used. See Section 2.4.

Figure2: Showing 25 grid cells would be helpful (maybe remove the notations "X" since the plot will be more compact.)

We will change the figure as suggested by both Referees.

We now show all 25 cells, and the actual used temporal lags as well.

L170: I suggest adding bias and the Nash-Sutcliffe model efficiency (NSE) (https://en.wikipedia.org/wiki/Nash–Sutcliffe_model_efficiency_coefficient) (or R2 score, coefficient of determination, common in machine learning applications) to the evaluation metrics, so it is easy to compare the results in this study with other papers.

See our response earlier: the second comment in Major comments of this document.

We did not include bias, but we included the R2 score.

L173: Use 1000 instead of 103 for easy reading.

We will change $10^3$ to 1000.

Changed as suggested.

L175: Hyperparameter tuning through a grid search or other techniques is a common procedure to obtain the optimal accuracy of a machine learning model. It is an essential step to create a fair game when benchmarking different models. Often hyperparameters are determined for each cross-validation fold (see Tramontana et al., 2016 for an example). Although it might be true that significant improvement in the model performance is not likely, it is important to include sufficient justification about your tuning process. For example, what was the search space of parameters? How many sets of parameters were evaluated?

We will add information about the details of the hyperparameter tuning, such as the searched space of parameters.

Determining hyperparameters separately for each cross-validation fold sounds potentially computationally too heavy, but we can consider it. Also, we can consider applying some automated approach, such as Bayesian optimization (https://scikit-optimize.github.io/stable/auto_examples/bayesian-optimization.html).

We managed to improve the tuning with the Bayesian optimization approach. The most significant change after implementation of the new tuning approach was found in the colsample_bynode and colsample_bytree parameters, which were reduced to 0.10.

L180: Another suggestion is to perform repetition experiments (e.g. 30 or 50 repeated experiments for each algorithm, each with a different random split, and random state in the models) to estimate uncertainties from randomness in the cross-validation split and model initializations. See Besnard et al. (2019) for an example. In this way, the model comparison is robust to algorithm and splitting randomness. Confidence intervals of RMSE/R2 can also be derived this way, instead of bootstrapping within the samples.

We will also consider this approach, which sounds promising. Most likely reducing the num_parallel_tree from 10 to 1 is necessary to accomplish the heavier computation, which will slightly deteriorate the skill of the GB model.

We managed to perform repetition experiments, but we had to limit the repeats to 8. Still, the total actual repetition number was 40 = 8 repeats * 5 folds, which is considerably better than 5 folds only.

**Results**
L189: Do you mean 1,000 samples?

Yes – formatting of the number ($10^3$) was lost at some stage of the text processing.

Figure 4: 1000 bootstrap samples?

Yes – formatting of the number ($10^3$) was lost at some stage of the text processing.

L222-224 (Figure 4.): The variation of accuracy between years can also be related to the random split of years during cross-validation. For a test year, if years with similar climate conditions are in the training set, the testing accuracy is likely higher than otherwise. To this end, the repeated model runs would help eliminate this effect.

We can consider this approach.

We changed the figure as suggested. It turned out that cross-validation variability is much smaller than the variability found via Bootstrapping.

L232: do you mean "sub-sampling" here?

Yes. We will include the word "sub-sampling" in this sentence.

The sentence was changed as suggested.

L230-240: The description of methods should be in Section 2, and here you may present the results.

We will move the description to Section 2.

The description was moved to Section 2.7.

L265: I suggest placing Figure A1-3 to the main text, and Figure 6 can be presented in Appendix. Figure A1-3 summarizes the importance of individual ERA5 variables, different spatial grid cells, and information from different temporal windows respectively. They are easier to interpret and provide a clearer comparison than Figure 6.

We agree that Figures A1–A3 could be included in the main text, and Figure 6 could be moved to Appendix instead.

We replaced Figure 6 with the Figures A1-A3, and removed Appendix completely.

L269: It is interesting but somewhat surprising that the nearest grid cell is not the most important in the model. Further investigation and explanation would be needed here. What is the size of the tower footprint? How heterogeneous is this area? Is the tower close to cell 9, which may have a similar plant composition as the tower footprint? Is this related to lateral flows? What is the dominant wind direction?

See Figure 2 and its explanation in the comments of Referee #1, showing the typical, scattered landscape of the nearest grid cell of Hyytiälä. The surrounding cells are quite similar. Mostly forest covered, lots of lakes, and when zoomed closer, agricultural land is quite dominant as well. The dominant wind direction is from the South-West.

Considering the 12-hourly 4DVar approach of the assimilation of ERA5 (https://www.ecmwf.int/en/about/media-centre/news/2017/20-years-4d-var-better-forecasts-through-better-use-observations), the relatively large share of derived or simulated (not assimilated) variables, and the sparseness of the observations, it is not perhaps that surprising that there are uncertainties both in space and in time of the reanalysis. We believe that tree-based methods can take into account these biases and weight the different cells so that NEE variability can be optimally modeled based on the combination of the data from different cells.

Note also the spatial coarseness of the ERA5 data that we used. We are planning to redownload the data in the original, denser resolution. How much this will change the results is yet to be seen.

The tower footprint is about 125 000 m2, which is now mentioned in the text. We have included discussion about the potential uncertainties of the reanalysis and their effects on the modeling in different parts of the text (see Section 2.3; Discussion).

L282-283: It is interesting that sensible heat and soil temperature alone could explain 90% of the variance in NEE. Is this for the 6-hourly or weekly model? This could be because diurnal and seasonal cycles dominate the temporal dynamics of NEE. Could you please provide more information on this analysis? For example, provide a figure like the heatmaps in Figure 4 to show if the accuracy of interannual variabilities drops when using only two variables.

The result is for the 6-hourly model. It is indeed likely that the temporal cycles can explain the good result: those two variables might be sufficient to describe the cycles. Additionally, in this time resolution the unexplained (small-scale) variability is likely to be small, as it has been smoothed out by the temporal averaging. Both these reasons probably enhance the skill metrics. This might be an important explanation for the skill of the model, which is seemingly high compared to other single-site studies, typically using either denser (1 min, 30 min, 1 hour) or lower (eg. 1 month) time resolution.

We can plot the heatmaps as suggested, and either include them in the article text or in the Appendix if interesting results can be seen from them. We will also make clear the effect of the selected temporal time resolution on the results.

Unfortunately, we did not have time to perform this additional experiment.

**Discussion and conclusions**
L324: By "exclude", do you mean that the redundant variables have low feature importance? It might be misleading to say the model excludes a variable.

We will reformulate the wording of the sentence. It is true that even though some variables might get near zero feature importances, and thus do not effectively participate in the prediction, they still can be found at least in some of the trees.

We changed the sentence to be less misleading.